

# Lightning occurrences and intensity over the Indian region:
# Long-term trends and future projections
Rohit Chakraborty[1], Arindam Chakraborty[1,2], Ghouse Basha[3] and Madineni Venkat Ratnam[3]
[1] Divecha Centre for Climate Change, Indian Institute of Science, India
[2] Centre for Atmospheric and Oceanic Studies, Indian Institute of Science, India
[3] National Atmospheric Research Laboratory, India
*Correspondence to:* Rohit Chakraborty (rohitchakrab@iisc.ac.in)
**Abstract**
Lightning activities constitute the major destructive component of thunderstorms over India. Hence,
understanding the long-term variabilities of lightning occurrence and intensity and their inter-relation with various
causative factors is required. Long-term (1998-2014) Tropical Rainfall Measuring Mission (TRMM) satellite-
based lightning observations depict the most abundant lightning occurrences along the Himalayan foothills, the
Indo-Gangetic plains and coastal regions, while the intensity of these lightning strikes are found to be strongest
along the coastal regions and Bay of Bengal. In addition, both the lightning properties show a very strong
intensification (~1-2.5% annually) across all Indian regions during 1998-2014 with the maximum trends along
the coasts. Accordingly, a detailed statistical dominance analysis is performed which reveals total column water
vapor (TCWV) to be the dominant factor behind the intensification in lightning events, while instability, measured
by the convective available potential energy (CAPE), and aerosols optical depth (AOD) jointly control the
lightning frequency trends. An increase in surface temperatures has led to enhanced instability hence stronger
moisture transport to the upper troposphere lower stratosphere regions especially in the along the coasts. This
transported moisture helps deplete the ozone concentration leading to reduced temperatures and elevated
equilibrium levels which finally results in stronger and more abundant lightning events as also evidenced from
the trend analysis. Consequently, the relationship between lightning and its causative factors have been expressed
in form of multi-linear regression equations which are then employed on multiple global circulation models
(GCM) to understand the long-term impact of urbanization on lightning over a period of 1950-2100. The analysis
reveals a uniform increase in lightning occurrences, and intensity from both urbanization scenarios; however, an
accelerated growth is observed in the RCP8.5 projections after the year 2050 as also observed from the surface
warming trends. As a result, lightning frequency and intensity values across the Indian region are expected to
increase alarmingly by ~10-25% and 15-50%, respectively, by the end of this century with highest risks along the
coasts and hence it requires immediate attention from policy makers.
*Keywords:* Lightning; occurrences; intensity; CAPE; TCWV; AOD; GCM.

**1. Introduction**
Intense thunderstorm events form a very common climatic feature over the Indian subcontinent. These phenomena
are generally accompanied by widespread lightning, wind gusts and heavy rainfall which induce various
socioeconomic hazards. However, among all these by products, lightning occurrences have been found to cause



the greatest damage to life with a death toll of more than 2500 every year since the last two decades (Livemint,
2000). In addition, the recent years have witnessed some most severe lightning calamities as per available records
claiming more than 100 lives on 25 June 2020 (Washington Post, 2020).
Over the tropics, the non-inductive (collision based) charging interaction between ice crystals and graupel
particles is found to be the major factor behind the evolution of lightning events during typical thunderstorms
(Takahashi, 1978; Mansell and Ziegler, 2013). According, in this mechanism, the magnitude of charge generated
per collision depends on the relative velocity of the colliding particles, the hydrometeor concentration of graupel
and ice and their corresponding size distributions (Shi et al., 2015) and these in-turn are controlled by the
atmospheric moisture content (total column water vapour), thermodynamic instability (convective available
potential energy) and the possibility of cloud nucleation from aerosols. Additionally, Kumar and Kamra (2012)
suggested that orographic lifting also has good influence on lightning but only in limited high-altitude regions of
Indian Subcontinent.
Lightning flashes are found to be significantly correlated with convective rain, total column water vapour
(TCWV), or surface relative humidity over both land and sea regions, according to previous studies (Price and
Federmesser, 2006; Siingh et al., 2011; Shi et al, 2018). This is because, higher humidity levels lead to stronger
hydrometeor concentration and updraft velocities, both of which contribute to intense lightning. Next, high values
of instability represented by convective available potential energy (CAPE) are essential for lifting the available
moisture with strong updrafts above the freezing level where they form ice and graupel particles which collide to
initiate charge separation and lightning and this has already been demonstrated both theoretically and statistically
in various previous research attempts (Galanki et al. 2015, Saha et al., 2017; Dewan et al. 2018).
Finally, coming to the impact of aerosols (AOD),  a study by Shi et al., (2020) reported that the lightning
flash rates are strongly correlated when AOD<1 due to the cloud/ice condensation nuclei formation characteristics
from sulphates (Jin et al., 2018), desert dust (Boose et al., 2019) or even sea salt aerosols (de Leeuw et al., 2011).
On the other hand, when AOD>1 then normally, larger concentrations of cloud condensation nuclei result in more
supercooled droplets leading to stronger lightning (Williams and Stanfill, 2002). However, excessively high
aerosol concentrations may also result in reduced cloud droplet size (Twomey et al., 1984) which reduces the
efficiency of non-inductive charging process. In addition, an excess of absorptive aerosols (such as black carbon)
warms the atmosphere and cools the surface (Kar et al., 2009; Talukdar et al., 2019) which further reduces the
CAPE and lightning. Hence the reported relationships between lightning and aerosols is still unclear so, further
studies are required to unravel it (van den Heever and Cotton, 2007).
Now from climatic point of view, a series of studies in recent years have shown that thunderstorm severity,
lightning and its various controlling factors have been increasing prominently in the recent decades and this has
been attributed to greenhouse emission induced surface warming effects both over India as well as across the
globe. A study by Shindell et al. (2006) depicted that a minimum 10% increase in lightning activity can be
expected due to every 1°C increase by global warming. Kandalgaonkar et al. (2005) suggested that a rise of 1°C
in surface temperature over India has led to a 20–40% enhancement in average lightning flash density. According
to Riemann-Campe et al. (2009) and Prein et al. (2017) a recent increase of temperature has led to a rise in moisture
ingress, consequently the frequency and severity of intense convective activities have shown a steep rise globally.
Over India, Murugavel et al. (2012) and Chakraborty et al. (2019) showed a systematic increase in CAPE
which was attributed to thermodynamic instability conditions and large-scale dynamics coupled with a decrease



in upper tropospheric temperatures during that period. Also, satellite measurements have shown a prominent
increase in aerosol concentrations over Asia due to the intense growth in urbanization and industrialization
(Massie et al., 2004). Consequently, a new set of research attempts have tried to express lightning and
thunderstorm severities in form of their causative factors which are employed on global climate models (GCMs)
to provide future projections of extreme events (Diffenbaugh et al., 2013). Romps et al (2014) expressed the
lightning flash rate in terms of the product between CAPE and precipitation rate which when implemented on 8
climate models revealed an increase in lightning by $12\pm5\%$ per degree Celsius of warming over the USA. Later,
a range of other proxies were also used over GCMs for lightning projections, but all of them also provided a
similar increase in lightning both globally as well as over the USA (Bannerjee et al., 2014; Romps, 2019).
However, as already hinted by Michalon et al., (1999), many years ago, most of the modelling attempts are
expected to fail in providing a holistic understanding about the changing lightning climatology; and interestingly
this is also found true at present as evidenced from major disagreements in magnitude of the projected trends
among all the above-mentioned studies.

With reference to the previous sections, it has been understood that first, a very few studies on lightning has

been done on the tropics and especially over the Indian region (Pereira et al., 2010) and none of the remaining
attempts have tried to depict the future projections of lightning. Second, all the above-mentioned studies have
utilized very poor resolution lightning datasets and hence did not provide a holistic mechanism behind the
climatological variations in lightning evolution. Finally, none of these attempts have tried to see the variation of
lightning radiance or intensity which is expected to be much more connected with the underlying physical
processes compared to the frequency values. Thus, in the present study, high resolution lightning datasets of
frequency and intensity are used over the Indian region and an attempt is made to identify the most dominant
factors affecting the spatial-temporal variabilities of the lightning properties in a complete manner. Finally, the
results of this dominance analysis are expressed in the form of a multi-variate regression analysis and subsequently
applied on multi-model GCM datasets to generate reliable future projections of lightning occurrences and intensity
over Indian for next few decades subject to various urbanization scenarios.

The main organization of the paper is explained as follows. A detailed illustration of the various datasets

used in this study are elaborated in the datasets section. Next comes the results section which is again divided into
four subsections. The first two subsections discuss the spatial distribution of lightning and its 17-year trends and
they also try to identify the dominant factors responsible for the multi-decadal changes in lightning. In the next
part, a probable physical mechanism is proposed to relate how the recently accelerated global warming trends can
modulate the climatic intensification and abundance of lightning. However, the final subsection tries to implement
the above-mentioned hypothesis on multiple long-term global climate model datasets to provide reliable future
projections of lightning intensity and occurrence subject to various degrees of urbanization. In the final section of
the manuscript, all the results have been summarized to produce a simplified picture which tries to quantify the
long-term impacts of global warming on lightning extremes over the Indian region for future policy makers.

**2. Datasets used**
Lightning observations for the present study are obtained from Lightning Imaging Sensors (LISs) onboard the
Tropical Rainfall Measuring Mission (TRMM) satellite which orbits the earth at 350 km elevation between 35°
N and 35°S at a rate of 16 orbits/day (Christian et al., 2003). These LISs can detect both intra-cloud and cloud-to-



ground discharges irrespective of day or night conditions with a flash detection efficiency of 73±11 % and 93 ±
4%, respectively (Boccippio et al., 2002). The lightning observations are done by monitoring illumination pulses
along the 777.4 nm atomic oxygen multiplet with a very fine spatial (5.5 km) and temporal (2 ms) resolution.
Every time an illumination pulse registers intensity greater than the predefined background noise level, it is
considered as a separate lightning event after which all such events occurring within an integration time of 330
ms are collectively considered as a single lightning flash. The view times of these flashes are also recorded
separately for obtaining the monthly flash rate climatological datasets. Here it may be noted that most of the past
researches have used pre-processed monthly averaged lightning flash rates after 2.5° and 99-day smoothing of the
actual data which may have compromised the actual distribution of lightning properties in those cases. Hence, in
this study, the actual lightning observations from ~95800 satellite passes have been utilized during the total
availability period of 1998-2014. These lightning flash (/km2) datasets are compiled monthly and then averaged
annually while the lightning radiance values (J/m2/steradian/s$^{-1}$) are averaged directly on a yearly basis so that the
magnitudes of both these parameters remain along the same scale of 0-1 for simplicity in analysis. However, for
ease of region-wise analysis, the yearly time series of both these parameters has been analysed on a fixed grid
resolution of 1 degree.
Next, the gridded datasets of the causative meteorological factors namely CAPE and TCWV are utilized
from the Climate Forecast System Reanalysis (CFSR) developed by NOAA's National Centre for Environmental
Prediction (NCEP) (http://nomadl.ncep.noaa.gov/ncep-data/index.html) datasets (Kalnay et al., 1996) as provided
by the ESRL PSD during the period 1948-2014. These datasets are provided at a coarse spatial resolution of 2.0°
× 2.0° for CAPE and 2.5° × 2.5° for TCWV, hence it had to be interpolated to 1° resolution and averaged yearly
for subsequent analysis. Secondly, the datasets of aerosol content (AOD) over the maximum availability period
of 2000-2014 are obtained from Level-3 (L3) MODIS TERRA Atmosphere Monthly Global Product MOD08_M3
at 1X1 grid resolution (Platnick et al., 2015) over the Indian sub-continent. Further details can be obtained
herewith (http://gdata1.sci.gsfc.nasa.gov/daac-bin/G3/gui.cgi?instance_id=aerosol_monthly ). Gridded altitude
datasets at 0.25° resolution are taken from GMTED2010 global digital elevation model under TEMIS project
(Danielsen and Gesch, 2011). Finally, the future projections of lightning properties for various urbanization
scenarios are derived using gridded datasets of the temperature, humidity and ozone profiles with aerosol optical
depth at 550 nm from 11 general circulation models (GCMs) in the Coupled Model Inter-comparison Project
(CMIP5) archive (website: http://cmip-pcmdi.llnl.gov/cmip5/ ) during 1950-2100. Further details of these datasets
have been provided in Taylor et al. (2012) and also in subsequent sections of this study.

**3. Results and Discussion**
**3.1. Spatial distribution of lightning properties**
The climatological average of lightning frequency shown in Figure 1(a) depicts much higher values over the land
regions compared to Arabian Sea (AS) and Bay of Bengal (BoB). This is due to occurrence of stronger sensible
heat fluxes over the land regions resulting in stronger updrafts, and hence more lightning (Kumar and Kamra
(2012). The highest magnitude is observed along the foothills of Himalayas (72-95°E) which implies the effect of
orographic convection on lightning events. In these regions, the high values of lightning flashes are found
associated with the occurrence of a mountain breeze front during the afternoon hours (Boeck et al., 1999).  The





secondary spatial maximum of lightning is observed along the coastline which can be attributed to widespread
moisture advection from the adjoining seas (Kumar and Kamra, 2012).

The BoB experiences moderately high lightning frequencies due to high seas surface temperatures (SSTs)

(above a critical threshold of 28ºC according to Gadgil et al. 1984) which lead to frequent cyclonic storms and
lows in this region. However, AS experience lower lightning frequencies due to lower SSTs in this region (Kumar
and Kamra, 2012). Next, low lightning frequencies are observed along the peninsula due to reduced moisture
supply as it is geographical bounded by mountainous terrain along the coasts from both sides. In addition,
moderately high lightning frequencies are observed along the Indo-Gangetic plains (IGP) which can be due to the
complex interactions among the moderate moisture supply from BoB, local instability and the CCN effects from
transported (Boose et al., 2019) and emitted aerosols. Finally, very low values of lightning occurrences are seen
over West Central India (WCI) which can be due to dearth of moisture supply despite the contribution from
transported dust aerosols here.

Contrary to the occurrence climatology, the lightning radiance values (Fig. 1b) are much lower over majority

of land regions which is solely because of higher values of moisture content over the coastal regions leading to
more graupel (ice and hail particles concentrations above the freezing layer) as shown in Murugavel et al., 2012.
Yet, the maximum values of radiance are observed exactly along the coastline regions which reduce gradually as
one move further into the seas. This indicates the importance of thermal land-sea contrast which results in strong
moisture advection from both land and sea breezes along the land-sea boundary (Pielke, 1974) resulting in more
hydrometeors hence largest radiance. However, the presence of giant CCN marine aerosols particularly in AS (de
Leeuw et al., 2011) may also act as a secondary factor. Nevertheless, AS still experiences lower lightning radiance
than BoB probably due to its local meteorological factors as described earlier. Next, the lightning radiance values
are found distinctly lesser over PI, WCI and HIM as they receive much less moisture than the rest of India. In
Addition, a secondary maximum of radiances is observed over the IGP due to moderate moisture supply and
aerosols which can act as potential ICN/CCN thereby increasing the number of colliding hydrometeors for more
lightning radiance.

So, based on these spatial distributions of lightning occurrence and intensity, a group of 7 regions are

proposed for further analysis as depicted in Fig1d. The two coastlines share high lightning occurrences as well as
intensity hence are referred as Coasts. Second comes PI as a landlocked region between the Coasts with moderate
values of both lightning properties. Region 3 and 4 are taken for both sea regions namely BoB and AS as their
response to lightning occurrences and radiances are quite different. Next, the Himalayan foothills are considered
to observe the effect of orographic convection on lightning properties independently. Then, IGP is selected since
it experiences quite high lightning occurrences and intensity due to complex aerosol, instability and moisture
interactions. Finally, WCI a remote inland region is considered as it experiences lower lightning properties due to
meagre moisture supply despite an important contribution from land heating and aerosols.

The climatologically averaged distributions of lightning frequency and intensity along with 4 most potential

factors influencing them (TCWV, CAPE, AOD and Altitude) for different regions are shown in Fig. S1. For
lightning frequency, the mean and percentiles are highest in HIM and lowest in PI and AS. However, the extreme
values for the Coasts and WCI are found to be higher than HIM because of frequent cyclonic and low-pressure
systems prevalent in these regions (Fig S1a). The Lowest lightning frequency is observed over AS. The radiance
and CAPE follow similar variability i.e., higher values are observed over BoB and Coasts due to more moisture





availability and thunderstorm occurrences while the other regions experience low to moderate values (Fig. S1b).
Yet a lot of extremes are observed in IGP and WCI which implies that the occurrence of surface heating lead to
higher cloud base heights and more ice-phase hydrometeors; hence more lightning in these regions as also
supported from previous studies (Price, 2009; Shindell et al., 2006).

The total moisture content (TCWV) is found to be highest along the Coasts and adjoining sea regions as
from lightning occurrences and intensity. Incidentally, the remote inland locations like PI and WCI receive much
reduced moisture supply which is the primary reason behind the lower lightning radiances in those regions. Next,
the IGP and WCI regions show highest values of AOD due to dust transport from Thar and Sahara deserts along
with large scale emission from various anthropogenic activities which have very complex impacts on convection.
Likewise, the Coasts also receive moderate aerosol supply from the adjoining seas as already described before
which further support lightning formation. The HIM region depicts the highest altitude variation among all the
regions, of which a significant fraction is present above 4 km height thereby supporting widespread orographic
lifting induced lightning activities. Next, PI and IGP exhibit an infinitesimally small altitude variation (<500 m)
which may not be sufficient to support any orographic convection. Whereas in WCI, the altitude ranges are
considerably higher which in turn provides a small but vital contribution towards lightning as also supported from
various past research attempts (Barros et al., 2004).

### 3.2. Temporal variation of lightning properties

### 3.2.1. Long-term trends in lightning frequency and intensity over the Indian region

The 17-year time series variations of mean annual lightning frequency and radiance are depicted for seven regions
and entire India in Fig. S2 along with their respective standard error values. Robust-fit regression analysis is
employed to study whether there are any statistically significant trends in the lightning frequency and radiance.
At the same time, a detailed description regarding the yearly % trends of lightning properties and their controlling
factors (TCWV, CAPE and AOD) are depicted along with their corresponding correlation coefficients in Fig. 2.
The trends of lightning frequency are found to be the highest over the Coasts, BoB and AS (with a total increase
of ~30%) with reasonably high correlation coefficient values which again can be attributed to an increase of both
moisture content and instability in these regions. However, the other regions depict much weaker values of both
these quantities. Here it is interesting to note that IGP depicts much weaker lightning trend which may be due to
the complex aerosol interactions as explained before. But in total, India has faced ~25% increase in lightning
frequency (with very high correlation values) in these 17 years which is alarming and hence will be discussed in
detail later in the study.

The mean annual lightning radiances show gradually increasing trends (with a total increase of ~20%) in
almost all regions. However, the magnitudes of % trends as well as the correlation coefficients are much lesser
compared to lightning frequency which implies that average radiance may not be a suitable parameter to
investigate the future variations in lightning. It is known that, TRMM observes both cloud-cloud and cloud-ground
types of lightning strikes together and out of the total, only the strongest 10% of the total strikes are intense enough
to reach the ground in the tropics (Uman, 1986) and cause immense damage to life and property (Holle et al.,
2019). Now, since it is more important to understand the trends of these extreme cases only, hence the regional
trends of 90[th] percentile of lightning radiances is examined (Fig.2a). The trends depict a very prominent all-India
trend of ~30% with higher correlation values compared to mean lightning radiance (Fig.2b). Additionally, the



coastal and sea regions depict much higher trends (>40%) than the rest of India which is extremely alarming for
policy makers at present. However, HIM has not shown any change in lightning radiance which may due to the
marginal increase in TCWV and CAPE there.

Further, it is investigated whether the proposed 90th percentile of radiance also agrees well with the

distribution of mean radiances over every Indian region; hence the corresponding values of both the quantities for
a total span of 68 seasons in 17 years are shown in Fig. S3. The figure depicts that the south Indian regions namely:
BoB, Coasts, PI and AS (with stronger maritime influence) exhibit prominent correlation values between the two
groups and the average ratio between the two is ~1.2; with a minimal amount of spreading which indicates that
there are no external factors affecting the average radiance distribution. But the northern inland zones depict a
prominent scattering between the groups (especially in IGP) and the ratio between them is also much higher (~1.4)
which indicates that some external factors such as aerosols may also exert an additional impact in intensifying the
radiance values well above the average radiance distributions. But overall, a good agreement is seen between the
lightning radiance groups thereby supporting the suitability of using p90 radiances only in the subsequent sections.

Next, the trends of TCWV (Fig. 2a) depict uniform trends (0.3% yearly or 5% in total) across all regions

with descent correlation values everywhere except HIM which are caused by prominent GHG induced global
warming in the recent decades as shown in previous studies. CAPE which represents the atmospheric instability
is the main reason for lightning evolution, hence this parameter also depicts strong interzonal variability like
lightning. The Coastal regions and seas experience much stronger increase of ~9% in 17 years implying more
thunderstorm activity in the present due to prominent global warming induced land sea thermal contrast. However,
the rest of the country exhibits a much weaker increase (~5%) and further, the trends in IGP and WCI are even
lower due to the complex aerosol effect. Finally, AOD is found to increase most significantly throughout India
compared to the other parameters, however the south Indian maritime regions experience a much lower rise of
~25% compared to deep inland regions (WCI and IGP) with ~40% increase indicating that aerosols may have
more dominant role in modulating the trends of lightning properties only in the continental North Indian regions.

**3.2.2. Investigation of the dominant factors affecting the lightning trends over India**

The previous sections depict a series of spatially varying complex interactions among TCWV, CAPE and AOD
which resulted in an increasing trend in both lightning properties over the Indian region. Now, to identify the most
dominant factors affecting the trends of lightning, a clustering analysis is done for each Indian region. Hence the
datasets of lightning are taken for 15 years span (as AOD data is absent before 2000) and then they are sorted into
three clusters based on magnitude. The mean and deviation of these clusters are depicted in Fig. S4 and Fig. S5
with respect to corresponding values of TCWV, CAPE and AOD. Factors having dominant influence are
identified as those where the parameter mean increases sharply with the clusters with minimal mutual overlapping.

The analysis revealed that CAPE and AOD depict good clustering for occurrences while TCWV shows the

best results for radiances over Coast and BoB. AS also behaves similarly, but in case of occurrences, AOD fairs
slightly better than the others implying a dominant contribution from CCN forming marine aerosol transport in
this region. PI experiences lesser moisture ingress (due to its inland location) hence it relies more on dry
convections which makes CAPE the major governing element for both the lightning properties. No single factor
is observed to be dominant factor for lightning frequency in HIM as it mainly relies on orographic convection
processes as already explained in preceding sections. But for radiance, TCWV still remains a dominant factor
(Fig. S5). Next in case of IGP, CAPE exhibits fair linear clustering in occurrence while TCWV remains



dependable for intensity. WCI behaves similar to PI but, here a secondary influence of AOD is also observed
indicating the possible impact of transported dust (acting as ICN) which catalyses the formation of ice phase
hydrometeors leading to more frequent and stronger lightning events.
Next, the clustering analysis is performed over the entire Indian region and is shown in Fig. 3. Based on
clustering and correlation analysis for lightning frequencies, CAPE emerges as the dominant factor followed by
AOD. This can be supported theoretically as the generation of lightning events only requires the availability of
ice phase hydrometeors above the freezing level which is achieved mainly by the lifting mechanism due to CAPE
followed by the aerosol CN effect. On the other hand, in case of p90 radiance, based on clustering and correlation
coefficients TCWV emerges as the single dominant feature behind the strong rise in lighting radiances all over
India. This result can also be explained theoretically as the inductive/non-inductive charging density responsible
for lightning radiance is far more dependent on the local hydrometeor concentrations (arising from moisture
abundance) compared to their relative vertical velocities (controlled by CAPE). Now interestingly, AOD has
consistently maintained a complex secondary impact on both the lightning properties depending its tendency to
either favour lightning (by creating more ICN/CCN formation due to dust or sulphate aerosols) or negatively by
inducing cloud burn-off effect(due to black carbon aerosols) as already discussed in preceding sections. So, a
detailed study needs to be done to untangle the AOD effect in aerosol sensitive zones like IGP and WCI.
Now, the temporal dominance analysis is repeated quantitatively using a multi-linear regression analysis. In
this step, two equations are hypothesized where lightning frequency and radiance are expressed separately as a
multi-linear addition of all three controlling factors. The proposed equations can be expressed as:
Lightning frequency = a1*TCWV + b1*CAPE + c1*AOD                    (1)
& Lightning intensity = a2*TCWV + b2*CAPE + c2*AOD                    (2)
Here, a, b and c represent the corresponding Multiple linear regression (MLR) coefficients for the three
factors and the numbers 1 and 2 stands for lightning frequency and radiance, respectively. The corresponding
variation of these MLR coefficients is shown in Fig. 4. CAPE acts as most dominating factors in all the regions
expect over AS where AOD influence is very high. AOD is the second most controlling parameter in lighting
frequency (except over BOB and HIM). For radiances, the TCWV is the dominant factor in most regions (except
over PI and WCI). The reasons for this were discussed in previous section. Again, similar to AS, AOD plays the
most significant role in modulating both the lightning properties over IGP due to the role of complex aerosol-
cloud interactions.
Finally, over the Indian region, TCWV arises as the dominant parameter controlling the climatic trends of
radiances; however, for occurrence it is not so simple. Though CAPE manages to be the principal factor, yet the
relative contributions of AOD followed by TCWV cannot be neglected. Next, the applicability of the proposed
MLR equations for long-term studies are validated by showing the ratio between regressed and observed lightning
properties (Fig4 i & j). In case of occurrences, the ratio between the two is not perfect and a small overestimation
of ~5% is observed hence this bias has been corrected before using it in the coming sections. Whereas, the
regressed values of radiance match perfectly with observations. Henceforth, these MLR equations have been
utilized for deriving the reliable long-term projections of lightning properties in subsequent sections.
**3.3 Physical mechanisms driving the increasing trends in lightning properties**
In this section, the physical processes responsible for the increase in lightning occurrences and intensity over the
Indian region will be discussed. Recent studies showed a prominent increase in aerosols and GHG emissions over



the Coasts, IGP and WCI as seen from the very strong increase in AOD in the recent years. This phenomenon resulted in widespread surface and atmospheric warming (Basha et al. 2017) and consequently a stronger surface evaporation and moisture production. In addition, many recent research attempts have reported a net increase in the Hadley cell and Brewer–Dobson circulation strength (Liu et al., 2012; Fu et al., 2015), which also assists in additional moisture supply. Consequently, the increased moisture in the atmosphere further accelerated the warming effect and TCWV growth in forms of a positive feedback (IPCC, 2007) primarily in the Coastal and neighbouring sea regions like BoB and AS. However, the increased moisture supply in IGP or PI is mainly due to the enhanced land-sea thermal contrast effect (due to GHG and aerosol emissions) which intensifies the moisture converges in these regions.

To explain how thermodynamic instability or CAPE has been increasing recently, a previous study by Chakraborty et al. (2019) is referred where long-term multi-station radiosonde observations depicted strong increasing trends in CAPE and TCWV all over Indian region with the maximum values along the coasts and surrounding inland regions.  However out of the total column, the percentage trends in both instability and moisture integrals (CAPE or TCWV) are found to be particularly higher above 300 hPa pressure levels which can be associated with a gradually ascending level of neutral buoyancy (LNB/EL) during this period. Now as the EL comes very close to the 100 hPa level during intense convective events, hence an observed cooling at its immediate surroundings (135-95 hPa) is thought to be the main factor responsible for the EL ascent and CAPE increase in these regions. The main reason for considering this hypothesis is based on a study by Dhaka et al. (2010) where a very prominent anticorrelation was observed between the yearly average values of CAPE and their corresponding upper-tropospheric temperatures at 100 hPa.

It has been well documented in past studies that ozone molecules act as the primary heat source component at 100 hPa level (corresponding to the UTLS region) by absorbing the ultraviolet radiations (Mohankumar, 2008). Now, the multi-station radiosonde observations from Chakraborty et al. (2019) depicted a clear rise in specific humidity and a depletion in ozone mixing ratios at the same height range. These results were analogous with the findings from Forster et al. (2007) according to which the recent decades have experienced an upper tropospheric cooling due to a decrease in ozone concentration. Thus, a cooling trend at this height level can be explained by the theory that excess moisture pumped to this height by intense convections get disassociated photolytically by reactive oxygen atoms to produce two OH radicals which further decompose ozone to oxygen molecule and a reactive oxygen atom in the UTLS region (Guha et al. 2017) thereby continuing the process. Consequently, this feedback process would lead to a further ascent in EL and increase in CAPE; however, the magnitudes of the resultant CAPE intensification will be highest over the coasts and surrounding seas due to a stronger moisture advection in those regions.

Hence according to this hypothesis, the Coastal regions and seas experience more growth in TCWV and CAPE which lead to formation of more ice phase hydrometeors thereby promoting an accelerated rise in lightning radiance. On the other hand, larger CAPE favours more updraft velocities in the ascending particles which further increase the probability of hydrometeor collisions leading to an increased lightning frequency. However, an additional effect can also be cast by AOD by facilitating more CN formation (from dust, sulphate or sea salt aerosols) which will strengthen the above-mentioned physical mechanism thereby leading to a stronger increase in both lightning properties over the aerosol sensitive inland regions such as IGP, WCI and PI.

**3.4 Generation of reliable future projections of lightning frequency and intensity**



### 3.4.1 Selection of GCMs for future projection analysis

In this section, the MLR coefficients from previous sections are employed to provide reliable projections of future lightning activities over the Indian region. Accordingly, the datasets of CAPE, AOD and TCWV are utilized over a period of 150 years including first 55 years (1950-2005) from historical datasets and the rest (2006-2100) from two extreme future scenarios namely: RCP2.6 and 8.5. A set of 8 global climate models (GCM) of CMIP5 (depicted in Table S1) are selected for analysis as all of them commonly provide the monthly mean estimates of TCWV and AOD with daily profiles of temperature, humidity and ozone. The monthly average values of CAPE are then calculated from daily T and RH profiles using the parcel approximation technique as described in past research attempts (Chakraborty el al., 2018; Narendra Reddy et al., 2018). It may be noted that, the surface-based CAPE (SB-CAPE) calculation technique has been used to obtain the current CAPE values in this study since they measure the total buoyancy experienced by the parcel raised directly from the surface to any height of the atmosphere irrespective of the prevailing atmospheric conditions, seasonality or the region where it is being derived. Now, the datasets from each of these models are interpolated to a uniform 1X1 degree resolution after which their performances are tested by comparing the simulated CAPE, TCWV and AOD values with respect to NCEP NCAR reanalysis datasets. The results of this test are represented in terms of a Taylor diagram in Fig. 5.

The Taylor diagram results for TCWV, CAPE and AOD unanimously reveal that models ACCESS1.3 CSIRO MK3.6, MIROC5 and NOESM 1ME (represented as A, B, F and H in Fig.5.) depict good correlations over the Indian region along with lower std and rms values. In addition, the model derived monthly inputs are also validated against NCEP data for all seven regions in Fig. S6. The correlation coefficients for all regions commonly show that models ACCESS1.3 CSIRO MK3.6, MIROC5 and NOESM 1ME again show much better agreement with NCEP data. Hence these four models are considered further for lightning projection analysis. Next, the all-India averaged regressed lightning occurrences and intensity values obtained from the models during 2000-2014 are plotted against their corresponding observations to check the reliability of MLR analysis on the modelled data. The results depict a fair agreement between the two sets (r=0.76 in occurrences and 0.7 in radiance) in both the cases. However, a very prominent underestimation bias has been observed (~24% in occurrences and ~18% in radiances) which is probably because the modelled datasets are of much coarser resolution than the actual observations, hence they will always depict much lesser average or variability compared to the former. However, the inter comparisons between the modelled and observed lightning properties over all the regions commonly depict quite high values of correlation with an overall underestimation bias of 17-25% thereby supporting the reliability of MLR analysis. However, the underestimation biases obtained from inter-comparison tests must be added with the regressed climatic projections for both zonal and all-India cases to get the actual lightning trends in forthcoming sections.

### 3.4.2 Examination of the 150-year trends in various factors controlling lightning

The 150-year trends of various controlling factors associated with the climatic trends of lightning occurrences and intensity are shown in form of normalized % change per decade for all seven Indian regions in Fig. S7. The surface temperature trends are first considered as this parameter is closely associated with urbanization and GHG emissions and it also acts as the primary driver behind the CAPE and TCWV trends. A moderate rise in RCP2.6 is observed (~0.5% per decade) which represents an all-India warming by ~1.6°C in total while the RCP8.5 scenario exhibits an extremely severe warming of ~5°C in the Coasts, WCI, BoB and IGP which is also expected to cause a parallel increase in TCWV and CAPE in future. Next, at par with surface warming, TCWV exhibits a





moderate increase from RCP2.6 scenario but in case of RCP8.5, an alarming growth of ~40% is observed across
India (with the largest increase in the Coasts, BoB and AS) which will definitely lead to a parallel huge change in
extreme lightning radiances over the total span of 150 years.

In case of AOD, contrary to the extremely large increase of ~30-40% between 2000 and 2014 from MODIS,

a much smaller rise of only 20% is seen during a much larger span of 150 years. Now to understand the source of
this discrepancy, the 150-year time series of AOD is observed which reveals that the initially increasing trend of
aerosols reverses to a strong negative trend after 2020 which results in an overall weak positive trend. The sudden
decline in AOD can be explained by the fact that RCP2.6 scenarios are characterized by stringent control on GHG
emissions and aerosols after 2020. However, RCP8.5 scenarios exhibit a higher overall increasing trend
amounting to 60%. Now this improvement in AOD trend from the latter case is because after 2020 the AOD
values saturates and then it shows a weak negative trend implying minor aerosol emission restrictions in future;
hence, the net cancellation of trends does not happen here. Also, the net increase in AOD is moderate in the Coasts
but much higher in aerosol sensitive regions like IGP and WCI implying even a doubling of AOD in these regions
which again may cast some vital influence on the lightning frequency trends of these regions in future.

Next in accordance with the TCWV and temperature trends, CAPE and MLCAPE depict a 15% and 8%

increase in total from RCP2.6 scenarios. Also, the CAPE trends are much higher than in MLCAPE which indicate
the validity of the upper tropospheric intensification theory as explained earlier. Again, the trends in CAPE and
MLCAPE are the highest being over Coasts and surrounding regions due to maximized moisture availability as
also shown in the previous studies. However, the RCP8.5 scenario shows an alarmingly high total trends of ~50%
and 20% in CAPE and MLCAPE respectively due to intensified global warming and moisture availability with
the highest rise of ~60% over the Coasts and seas which implies the possibility of accelerated growth in lightning
occurrence in these zones. However, inland regions (IGP and WCI) still show moderately high CAPE trends (due
to strong surface heating and aerosol trends) which may also lead to stronger lightning frequencies there.

Next, the trends in EL pressure level show an expected depletion (implying an ascent in EL) from RCP2.6

scenario with values between 0.5-1 % hPa per decade. However, in RCP8.5, the trends are further enhanced with
a range of 1-2% per decade with the largest changes occurring in the Coasts and BoB followed by IGP and WCI
due to a stronger increase in CAPE and TCWV. Similarly, the T100 cooling trends experience exactly similar
behaviour as EL with ~1-degree cooling in Coastal regions from RCP2.6 scenario while in case of RCP8.5, a
drastic cooling of up to ~2ºC is observed in total which can highly invigorate the convective strengths leading to
much stronger lightning events in future. Next, in RCP2.6 case, SHUM at 100 hPa undergoes ~0.3% increase per
decade associated with a 1% decrease in ozone. Here it may be noted that the ozone depletion trends are much
higher than in SHUM only because the photolytic disassociation of a single water vapour molecule with reactive
oxygen atom produces two OH radicals which help in decomposition of two ozone molecules. However, using
RCP8.5 scenario, these phenomena gets further amplified where -0.6% per decade increase in SHUM and (-2%)
depletion in ozone is observed with highest magnitudes observed in the Coasts. Hence the results suggest that
under higher surface warming (RCP8.5 scenarios), CAPE and TCWV will increase by exactly same hypothesis
as shown in Section 3.3 which ultimately results in a very strong increase in lightning properties over India

Now, coexistent with the zonal decadal trends the all-India time series of all parameters are shown in Fig. 6.

The surface temperatures depict an increase by 2 and 4 degrees, while the TCWV and CAPE also rise by 10%,
50% and 20%, 40% respectively for the two pathways. However, it may be noted that the main difference in


TCWV trends between RCP2.6 or 8.5 scenarios mainly arises from its drastic increase in the latter case after 2050
which is again attributed to the accelerated global warming conditions experienced using RCP8.5 scenarios during
those decades. Next, the AOD follows a dampened increasing trend in both scenarios with the increase in latter
being slightly more prominent than the former. Now these dampened AOD trends are expected to reduce the net
growth in lightning occurrences (owing to AOD's prominent contribution in the lightning frequency MLR
coefficients), but such effects will not be discernible in the radiance trends as it depends primarily on TCWV only.
Now because of the CAPE and TCWV trends, EL has shown a prominent ascent coupled with strong UTLS
cooling and increased moistening and ozone depletion trends in both urbanization scenarios. However, the trends
in RCP8.5 scenario are consistently much stronger than the RCP2.6 case due to much stronger GHG induced
UTLS dynamics and CAPE intensification feedback effect. In addition, the main difference between the trends
from both scenarios is mostly prominent towards the end of 21$^{st}$ century as explained previously.
**3.4.3 Expected overall trends in lightning frequency and intensity**
The 150-year trends in lightning properties for all seven Indian zones are observed in Fig. S7. The lightning
occurrences depict an overall increasing trend of ~15-25% for the total 150-year span which after adjustment for
underestimation bias (20-27% as in Fig. S6) provides the actual trends to be 19-31%. However, this increase of
lightning occurrences is rather low compared to the 17-year trends from observations. Hence, the lightning
frequency time series for each zone is investigated separately which reveal that the lightning frequencies have
increased up to 2020 after which it gets saturated; nevertheless after 2050, it again started to increase up to 2100
thereby leading to much lower trend values (Fig. 6). However, this type of variation can be explained by the
secondary influence of AOD on lightning occurrences which also shows a dampened increase thereby
compensating the impact of increasing CAPE in totality. In addition, out of all regions, the strongest increase in
lightning frequency is mainly observed in the Coasts, BoB, PI, IGP and WCI which is primarily due to influence
of CAPE and moisture supply in the first two and due to dry surface heating and aerosol effect in the rest. However,
in RCP8.5 scenarios, a much larger trend values amounting to 29-41% are observed (after bias correction) which
is mainly attributed to the stronger increase in CAPE and TCWV and a weaker decline in AOD in this case.
However, the spatial distribution of the trends remains fairly like the RCP2.6 case.

The 150-year zonal trends in lightning radiance from RCP2.6 scenarios depict a prominent overall increasing

trend of 35-54% after zone-specific bias corrections. Here the radiance trends are found to be much higher than
in occurrences since the lightning radiance trends depend primarily on TCWV which also shows a prominent
increase across the 150-year span. However, the net radiances are still a bit lower than the expected trends from
observations due to the small declining trend contribution from aerosols. However, the RCP8.5 scenario depicts
a very alarming increase amounting to (56-97% after bias correction) which can be attributed to the stronger
increase in both TCWV and CAPE throughout 150 years coupled with a weaker decline in AOD. In addition, the
lightning radiance trends are found to be the strongest in the Coasts and BoB, due to the accelerated rise in TCWV
and CAPE while a slightly weaker trend is observed over IGP and WCI due to the compensating influence of
AOD in addition to the TCWV and CAPE trends.

Now, at par with the zone-wise decadal lightning trends, the all- India averaged time series of lightning

occurrences and intensity are depicted in Fig. 7. The lightning occurrences from RCP2.6 scenarios depict a weak
increasing trend amounting to 26% (after bias correction of 24%) over the 150-year time span. As already
explained, the weak trend observed is due to the cancellation between increasing trends of CAPE and TCWV





against a declining trend in AOD. But in case of RCP8.5, a considerable increase in lightning frequency amounting
to 35% is observed which is mainly due to a strong rise in CAPE and TCWV along with a weaker decline in AOD.
In case of extreme radiances, RCP2.6 scenario shows a moderate rise of ~45% throughout India (after a bias
correction of 18%). However, RCP8.5 scenario depict a much higher increase by ~73% which is due to the much
stronger rise in TCWV followed by CAPE with minimal contribution from AOD. In addition, an exponential rise
in both TCWV and CAPE are observed after 2060 because of excessive GHG emission induced global warming
thereby leading to the highest increase in lightning radiance over India after 2060.

Now finally, an attempt is made to estimate the net % increase in lightning properties starting from the

present (2010-2020) in order to describe the probable difference in lightning trends if two extreme GHG emission
policies are adopted. Under these circumstances, the lightning frequencies depict a weak rise of ~13% assuming
RCP2.6 scenario but this trend would increase to ~19% in case of RCP8.5 while in the coasts, the trends may be
slightly higher reaching a maximum of ~22% for the latter. Yet, these trend values are quite smaller and hence
can be avoided quite easily in the future. However, in case of extreme radiance, the minimum possible increase
considering stringent policy making decisions from RCP2.6 scenario is ~22% throughout India. But in absence of
such restrictions (RCP8.5) the overall increase in extreme radiance will be ~37% with respect to present. In
addition, due to the impact of stronger TCWV and CAPE trends, the Coastal regions can face even up to a ~50%
rise in extreme lightning intensities by 2100 which poses an extreme socio-economic threat and hence requires
immediate mitigation strategies from policy makers at present.

## 495     4. Summary and Conclusions

Lightning activities are considered as an essential by-product of thunderstorms and they pose the greatest damage
to life annually since the last few decades. However, only a few studies have been reported over India or globally
which attempted to understand the evolution and distribution of lightning processes in overall and thereby provide
a reliable estimate about the future projections of lightning. Hence, the present study, attempts to utilize high
resolution lightning observations to explain its socio-temporal variabilities over India and also to identify the most
dominant factors responsible for the evolution of such extremes. In addition, the proposed inter-relationships
between lightning and its causative factors namely: moisture, instability and aerosols are also implemented in
multi-model GCM datasets to derive reliable future projections of lightning properties over the Indian regions for
the 21$^{st}$ century. The main highlights obtained from the present study include:
1. The highest climatological average of lightning occurrences is observed along the Himalayan foothills,

followed by coastal regions and Indo-Gangetic plains which are mainly attributed to the influence of

orographic convection, moisture ingress (due to land-sea thermal contrast) and aerosol cloud interactions.

2. Annual average values of lightning radiances are the strongest along the coastal regions and surrounding

seas primarily due to the dominance of hydrometeor concentrations on the lightning charge density equations

caused by enhanced moisture availability in those regions.

3. During the period 1998-2014, lightning frequencies exhibit a strong growth of ~1-2 % annually across all

Indian regions with a strong inter-regional variability. However, the trend values are invariant and quite

lower in the average radiance trends. Therefore, the trends in 90$^{th}$ percentile radiances are estimated which

show prominent spatial variation with 2-2.5% increase annually along Coasts, BOB, PI and IGP. These zonal

diversities in the lightning trends are also supported by the corresponding CAPE and AOD trends.



4. The clustering and multi-linear regression dominance tests depict that over India as a whole, atmospheric moisture (TCWV) is the principle factor controlling the lightning extreme radiance trends, while instability (CAPE) and aerosols (AOD) jointly play a strong role behind lightning frequency variation. However, in the latter, the proposed inter-relationships are found to deviate from region to region due to complex aerosol-cloud interactions towards thunderstorm genesis and lightning evolution.

5. Results from previous research attempts are employed to explain the underlying physical mechanism of these trends which inferred that an increase in surface temperatures has led to higher instability and moisture transport to the UTLS regions. This moistening resulted in ozone depletion and cooling which further uplifted the equilibrium levels leading to stronger CAPE and more ice-phase collisions above the freezing level; and eventually this complex feedback procedure ultimately leads to stronger and much more lightning events. A schematic of these processes has been depicted in Fig. 8.

6. The above-mentioned hypothesis is found to be most prominent in the coasts and surrounding seas due to its high moisture abundance. However, in addition to these factors, this mechanism will also be further invigorated/inhibited based on the prevailing RCP scenarios and also depending on the region-specific aerosol input on convective processes as discussed before.

7. These observed inter-relationships are expressed in form of multi-linear regression equations and then implemented on 4 suitable GCMs out of 8 available models during 1950-2100. The resulting multi-decadal projections reveal prominent trends in surface temperatures, moisture, instability and subsequent ozone and moisture concentrations in UTLS as proposed. However, a difference in urbanization rates led to much sharper trends in all parameters particularly after 2050 in the RCP8.5 case.

8. Consequently, the regressed lightning projections also depict an increase in both occurrences and intensity. However, the increasing trends are consistently higher in the RCP8.5 case. In addition, the increase in lightning frequency is found to be much slower than in case of intensities due to the dominant impact of AOD trends which also show a comparative saturation or decrease after 2020. This can be attributed to a probable increase in GHG emission restrictions by policy makers in near future.

9. Finally, the net intensification in lightning properties by 2100 with respect to present values depict that number of occurrences would increase moderately by (10-17 % and 16-23%) for RCP2.6 and 8.5 scenarios. However, extreme lightning radiances will increase much faster by 16-27% and 32-50% in RCP2.6 and 8.5, throughout India.

10. In addition to the overall trends, certain regions like the coasts and surrounding seas are prone to be at the higher lightning risk in future since they show much stronger increasing trends of ~50% (for radiance) in absence of stringent GHG emission restrictions (as in RCP8.5) which is extremely alarming and hence should be immediately addressed by policy makers.

It may be noted that this is the first ever study to use high resolution observations of lightning radiance as well as frequency over the Indian region where a holistic inter-relation between lightning and its causative factors have been proposed, tested and then implemented over a set of GCMs so as to provide a set of future projections for both lightning properties till the end of this century. Now, the lightning projections laid out in this study can be considered as reliable for forthcoming research attempts since both the equations and the models have been repeatedly validated against observational datasets. Nevertheless, the projected lightning increase due to global warming in this study is found to be much lesser than that obtained by simultaneous studies over the United States



(as evidenced from Romps (2019)), the reason for which can be possibly attributed either to stronger urbanizations
conditions in those regions or the choice of lightning proxies and GCM datasets used in that study.
However, from a closer point of view, the present study still has certain shortcomings. The primary limitation
is that it tries to provide an overall explanation for lightning trends over the Indian region. However, for specific
regions of the country such as WCI, HIM or IGP, secondary mechanisms from orographic influence or aerosol
effects (both radiative or microphysical) can also play a stronger role on the lightning trends and hence this
requires another dedicated study to address these issues. Secondly, the observed trends may vary strongly when
observed for separate seasons which have been averted here to provide a more focussed investigation on the
climatic trends of lightning. Finally, this study provides an overall mechanism of lightning; however, this
procedure may not be followed for all types of thunderstorm events, hence in future a suite of numerical models
and observations are required to explain how individual lightning events may be impacted by the complex aerosol,
instability and moisture interactions within the cloud over various meteorological conditions and Indian locations.

**Data availability**
High resolution lightning datasets for the present study have been obtained from LIS archives of NASA Global
Hydrology Resource Centre DAAC, U.S.A. ([https://ghrc.nsstc.nasa.gov/lightning/data/data_lis_trmm.html](https://ghrc.nsstc.nasa.gov/lightning/data/data_lis_trmm.html) last
access: 16 December 2020). Gridded datasets of CAPE and TCWV are utilized from the CFSR reanalysis archives
developed by NCEP ([http://nomadl.ncep.noaa.gov/ncep-data/index.html](http://nomadl.ncep.noaa.gov/ncep-data/index.html) last access: 10 December 2020). The
datasets of AOD are utilized from Level-3 (L3) MODIS TERRA Atmosphere Monthly Global Product
MOD08_M3 ([http://gdata1.sci.gsfc.nasa.gov/daac-bin/G3/gui.cgi?instance_id=aerosol_monthly](http://gdata1.sci.gsfc.nasa.gov/daac-bin/G3/gui.cgi?instance_id=aerosol_monthly) last access: 10
December 2020). Finally, the future projections of lightning properties are derived from 11 general circulation
models (GCMs) in the Coupled Model Inter-comparison Project (CMIP5) archive (website: [http://cmip-](http://cmip-pcmdi.llnl.gov/cmip5/)
[pcmdi.llnl.gov/cmip5/](http://cmip-pcmdi.llnl.gov/cmip5/) last access: 1 December 2020).

**Author contributions**
RC performed complete analysis and wrote the first draft. MVR and SGB provided the initial concept and did the
main editing while AC contributed to supervision, discussion, and editing.

**Competing interests**
The authors declare that they have no conflict of interest.

**Acknowledgments**
One of the authors (Rohit Chakraborty) thanks the Institute of Eminence Grant and Department of Science and
Technology for providing support under C V Raman Post-Doctoral fellowship and INSPIRE Faculty Research
Grant. He also acknowledges the Indian Institute of Science, for providing necessary support for this work. AC
acknowledges funding from the National Monsoon Mission, Ministry of Earth Sciences, Govt of India.






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

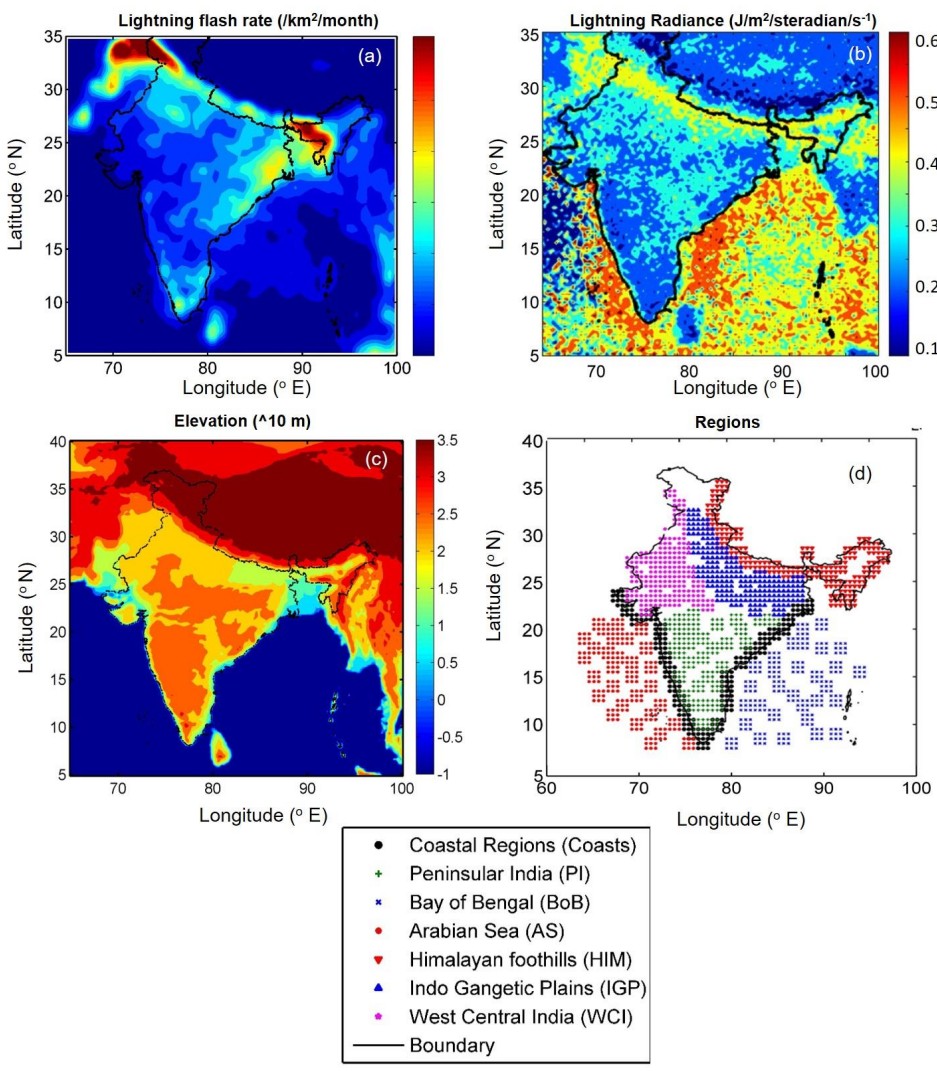


**Figure 1**: Climatological mean (a) lightning flash rate, (b) lightning radiance over India averaged during 1998-2014. (c)
Altitude above mean sea level in log10 scale (Seas denoted by -1) (d) Representation of seven regions used in the study











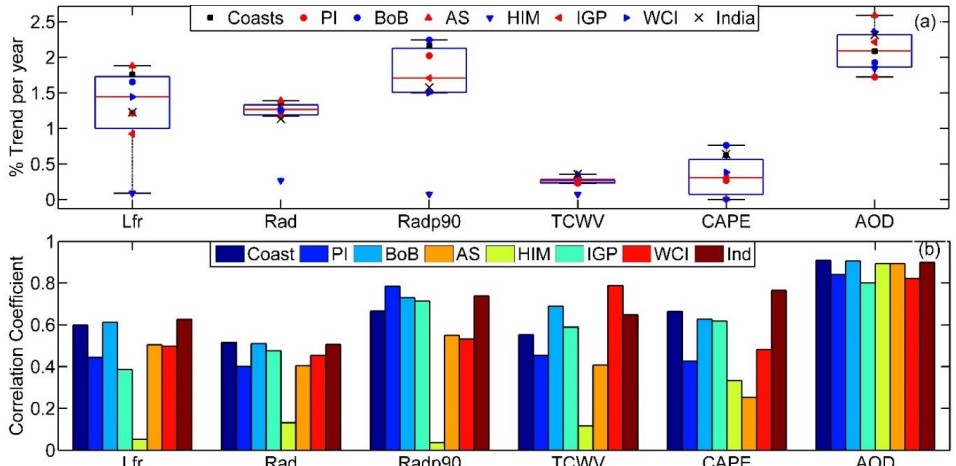


**Figure 2**: (a) % yearly trends in lightning flash rate (Lfr), Average radiance (Rad) and 90th percentile of radiance (Radp90)
with TCWV, CAPE and AOD over seven regions and all India datasets. (b) Correlation coefficients corresponding to this trend
values.
















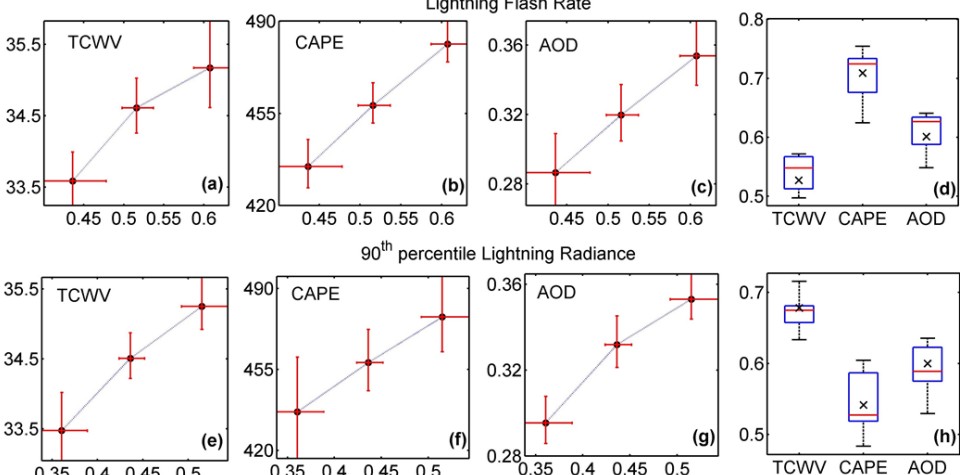

**Figure 3**. Temporal dominance cluster analysis results of lightning frequency and extreme radiances with respect to (a,e) TCWV, (b,f) CAPE and (c,g) AOD over entire Indian region, (d,h) distribution of correlation coefficient due to zonal clustering for both lightning parameters.





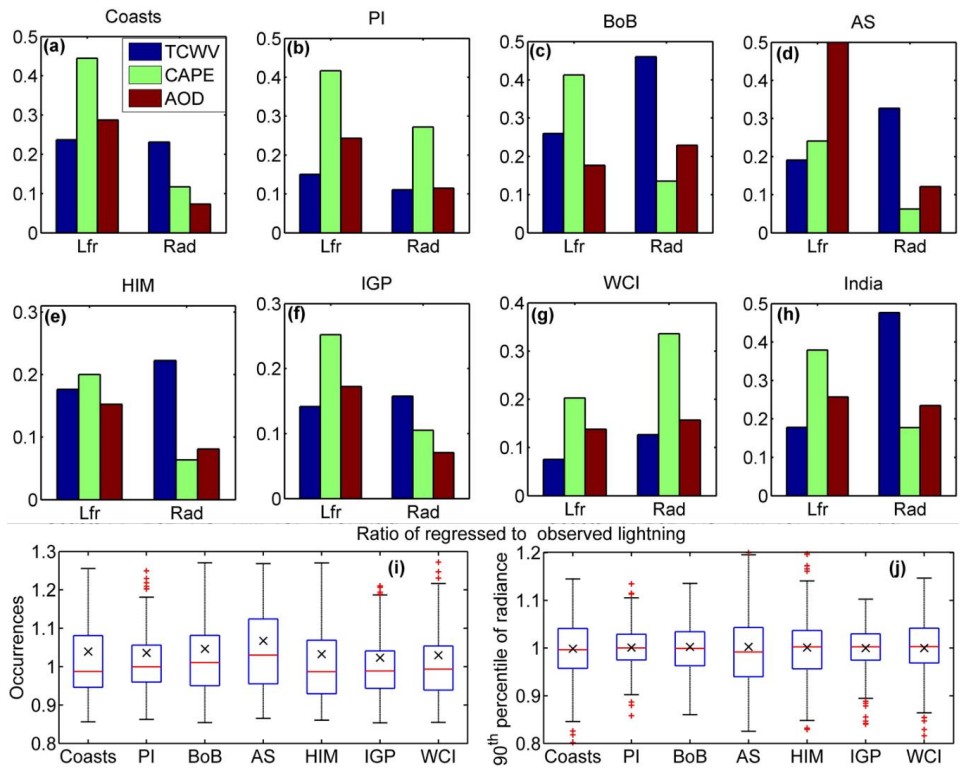

**Figure 4**: Temporal dominance analysis MLR coefficients of TCWV, CAPE and AOD for lightning frequency and radiance over (a) Coasts, (b) PI, (c) BoB, (d) AS, (e) HIM, (f) IGP, (g) WCI and (h) all India, (i,j) Zonal distribution of ratios between regressed and observed lightning frequency (Lfr) and 90th percentile of radiance (Radp90) over seven Indian regions.



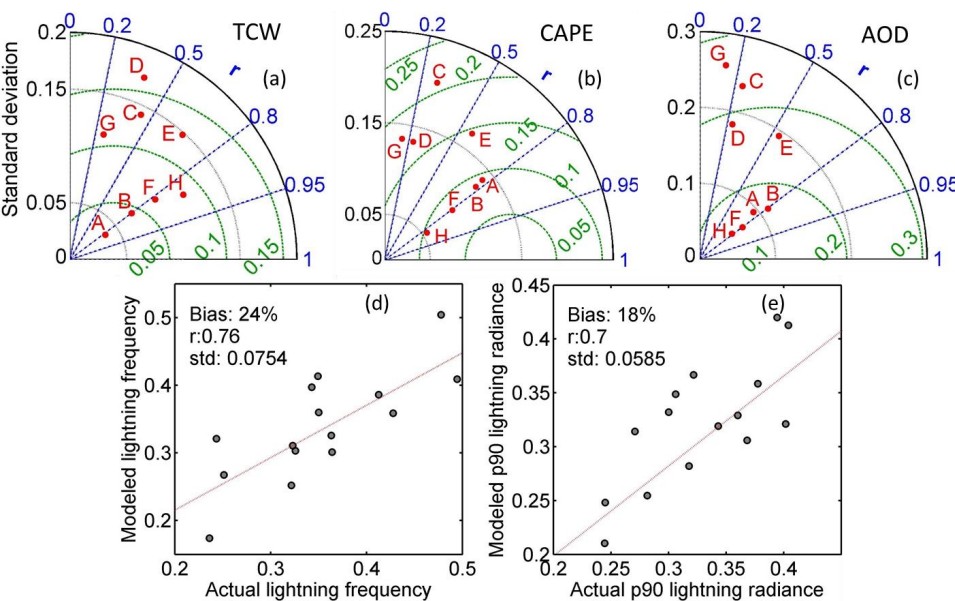

**Figure 5**: Taylor diagram representing the performance of 8 GCMs used in the study with respect to (a) TCWV, (b) CAPE and (c) AOD, (d,e) Covariation between regressed lightning properties from model mean with respect to observations for occurrences and intensity.






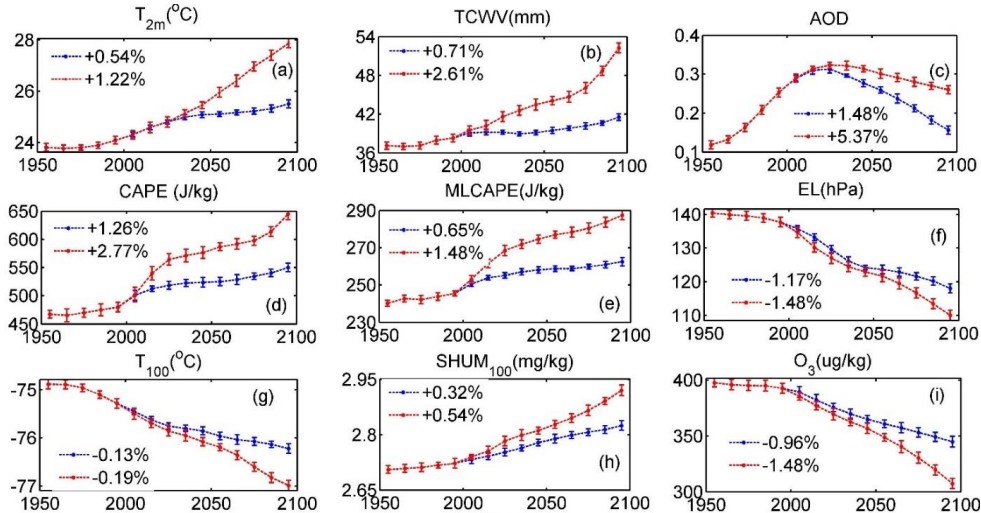

**Figure 6**: 150-year multi-model all-India average projections of various parameters using RCP 2.6 (blue) and RCP 8.5 (red)
scenarios for (a) 2 metre temperature, (b) TCWV, (c) AOD, (d) CAPE, (e) MLCAPE, (f) EL, (g) Temperature 100 hPa, (h)
Specific humidity, (i) Ozone mixing ratio same level.








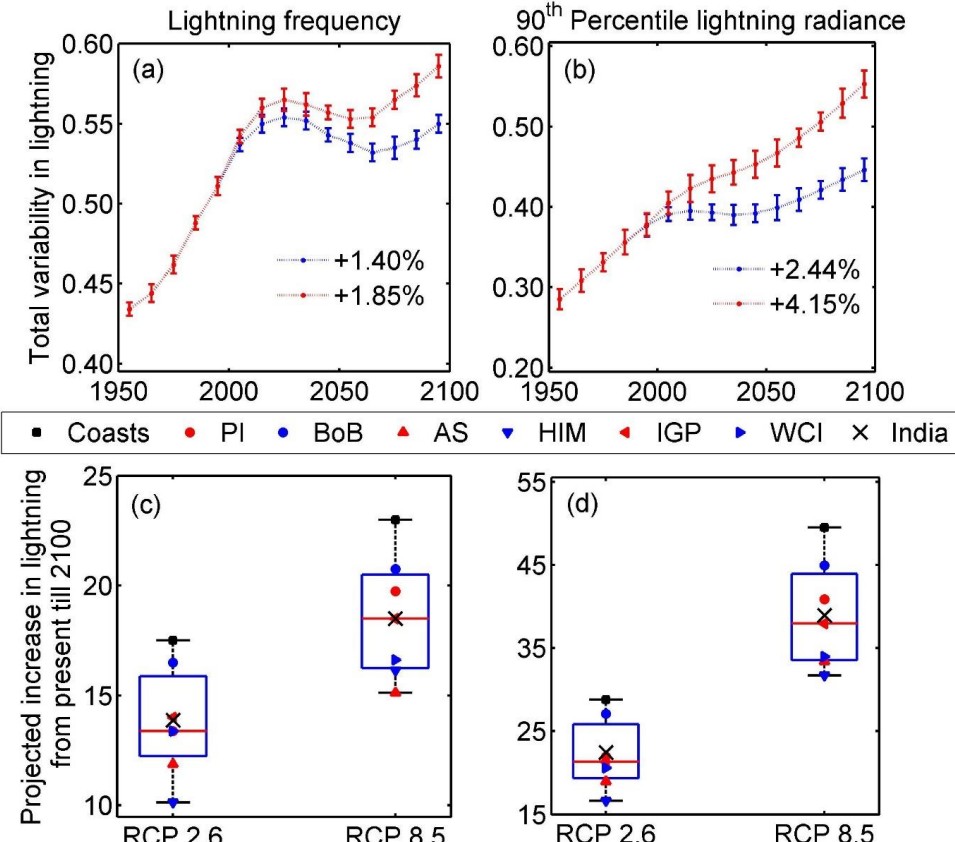


**Figure 7**: 150-year multi-model all-India average projections of (a) lightning occurrences and (b) 90th percentile radiance using RCP 2.6 and RCP 8.5 scenarios (c,d) Zonal distribution of trends in lightning occurrence and intensity between present decade (2011-2020) and 2091-2100

















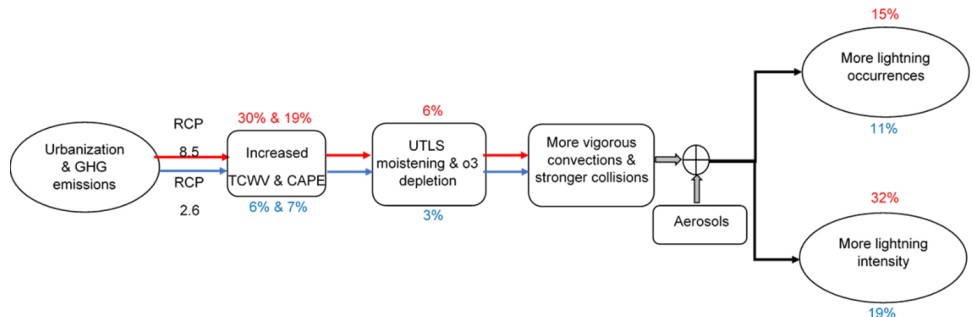

**Figure 8**: Proposed hypothesis to explain the long-term growth in lightning properties assuming RCP 2.6 and 8.5 over Indian
region