# Peer review of "Lightning occurrences and intensity over the Indian region"

_Atmospheric Chemistry and Physics, 2020_

## Author Comment (AC1)

**Replies to Reviewer #1**

General Comments: In general, the paper doesn't suffer from any major flaws, though it could benefit from better explanations and clarifications in both the text and in the figures. While I have noted that this manuscript should undergo major revisions, it is probably more like **in between** **minor and major**.

First of all, the authors wish to thank the reviewer for the comments/suggestions which significantly improved the content of the manuscript. The authors have addressed all the comments raised by the reviewer and incorporated them in the revised manuscript wherever required.

1. pg 2, line 11: Are the authors suggesting that at higher latitudes inductive mechanism is found to be the major factor? I am not aware of studies showing this. However, others, e.g., Saunders (1992), have claimed that the noninductive mechanism may be more responsible for early electrification and the inductive mechanism more so for later on during the thunderstorm development process. Please clarify.

Reply: The authors wish to clarify that they have never suggested about the dominance of inductive charging mechanisms in higher latitudes. In actuality, they have mostly focussed their research and literature survey towards the tropics from which they obtained sufficient number of references supporting the dominance of non-inductive mechanisms in lower latitudes. However, they have not done such an in-depth literature survey on lightning mechanisms over the temperate regions and hence are in no position to comment on this fact.

In this study, the authors have done a statistical dependence analysis to understand the evolution of all lightning activities from satellite observations irrespective of the thunderstorm life stage in which they occur. Now it is true that some studies have mentioned the possibility of separate lightning mechanisms being dominant in various thunderstorm phases. However, it still needs to be validated whether these hypotheses are at all valid during intense tropical convections also and this requires extensive observational and modelling research which has not been attempted yet. Consequently, a discussion on this topic is beyond the scope of this study; hence it has not been attempted.

2. pg 3, line 95-97: As the authors say, lightning radiance is generally not investigated. Some more background should be presented as to why this variable, which is probably less familiar to readers, is more connected with physical processes (e.g., lightning nitrogen oxide (LNOx) production), as well as why it is not correlated with flash frequency (along similar lines, why "radiance is far more dependent on local hydrometeor concentrations" than is flash rate, as mentioned on line 286). Lastly, provide some context as to why policymakers should care about radiance. From a possibly naive point of view, deaths are likely to be related to the number of flashes, regardless of the flash radiance.

Reply: At first, the authors wish to clarify that they have never tried to infer anywhere in this study that lightning frequency and radiance are uncorrelated because they basically determine the quality and quantity of the same entity. However, they have kept on emphasizing the dominant role of hydrometeor concentrations over radiance compared to flash rate. This is because, lightning radiance stands for the amount of charges generated either by graupel ice collision or by the inductive charging process during intense convective events. Now, a closer look into the charge production equations of lightning according to Shi et al. (2015) reveals the dominance of the size and concentrations of several hydrometeors like graupel or ice on the net charges produced compared to the other factors such as instability or aerosols. Now this dependence also makes sense because more the number of interacting hydrometeors, stronger the charges produced. But on the other hand, lightning frequency does not provide any idea whether it is a strong or a weak flash. This is because, weak flashes always tend to be far more abundant compared to the strong ones and further these weak events can happen even in

the presence of moderate instability and aerosols irrespective of the number of hydrometeors. Hence, this explains how hydrometeor concentrations matter more for lightning radiance than its frequency.

Now let us focus on why lightning radiance is important both physically as well as for policy makers. It is well known that lightning activities originate due to charge separation in mixed phase clouds, but they require a sufficient amount of electrostatic charge to shatter the insolation capacity of the atmosphere so that they can get neutralized by the earth surface. Now, a majority of lightning occurrences observed are generally not strong enough to reach the ground and hence they remain as intercloud lightning. Consequently, these events have no bearing in our climate or socio-economy. However according to some novel studies like Uman (1986) only 10-20% of the total lightning activities are strong enough to reach the ground and only these events can fix the atmospheric nitrogen to produce NOx which acts as a good fertilizer in the agricultural sector. But on the other hand, these strong lightning events also possess enough destructive power to cause significant damage to life and property. So, in a nutshell lightning frequency refers to the whole lot of lightning happening in the clouds out of which only a small fraction of events having higher radiance can reach the ground and hence be impactful towards human life. Thus, the intensity of lightning should arise as a more crucial factor compared to its frequency for present policy makers in order to prevent any impending chances of catastrophises in the future.

In view of reviewer's comment, the following has also been added to the revised version of manuscript.

*"While utilizing the lightning radiance measurements from satellite observations, it may be best suited to explain the importance of this data towards weather and climate as this attribute has never been extensively discussed in past research attempts unlike lightning frequency or flash rate. It is well known that lightning activities originate due to charge separation in mixed phase clouds, but they require a sufficient amount of electrostatic charge to shatter the insolation capacity of the atmosphere and descend to the earth surface thereby causing widespread damage to life and property. However, a majority of these lightning occurrences are not strong enough and hence they remain as intercloud lightning without any real impact on the climate or socio-economy. On the other hand, according to some novel studies like Uman (1986) only 10-20% of the total lightning activities remain strong enough to reach the ground thus inflicting widespread socio-economic impacts. Consequently, the climatological variation of lightning intensity or radiance also needs to be monitored very closely by present policy makers in order to prevent the chances of any impending catastrophises in future."*

3. pg 4, line 129: Are the units for "radiance" J/m2/steradian/micrometer? I believe the GOES-R GLM documentation notes that this quantity is actually a solid angle averaged spectral energy (and I think LIS outputs the same thing). Please double-check the units from the downloaded data files.

Reply: In view of the reviewer's suggestions, the authors have cross-checked with the downloaded data files of lightning radiance obtained from LIS onboard TRMM observations. However, they find that "**J/m2/steradian/micrometer**" is also being mentioned in those sources. Hence, this unit will continue to be used in the revised manuscript as well.

4. pg 4, line 152-153: I am not familiar with Kumar and Kamra (2012); could the land-sea frequency contrast be due to different aerosol amounts? Perhaps a map of AOD could be helpful for the reader.

Reply: The authors have shown the average spatial variation of TCWV, CAPE and AOD to clarify the confusion raised by the reviewer. Here the CAPE values have been shown in log scale for the sake of better representation. Now according to the reviewer, the prominent land-sea contrast in lightning frequency has been caused mainly due to aerosols and interestingly, this claim is also found to be supported from the spatial variations. However, this assumption fails drastically over mid-IGP and AS

regions which experience the least lightning frequencies despite high AOD values (as either the CAPE or TCWV value is not high there). Further, this hypothesis is also negated by the fact that BoB experiences very low AOD but it still gets more lightning than AS because it experiences higher CAPE and TCWV.

[Figure]

Figure: Average spatial variations of TCWV, CAPE and AOD over the Indian Subcontinent

Thus, the authors would humbly like to reaffirm that a complex suite of interactions among TCWV, CAPE and AOD jointly determines which region will experience stronger or frequent lightning events. Here it may be noted that this figure cannot be incorporated anywhere in the manuscript as a frequency distribution-based representation of these factors is already present as a main figure in older version.

5. pg 9, lines 330-333: The authors mention ozone depletion as the primary cause for increasing CAPE. Without thinking about ozone, CAPE would be expected to increase with global warming due to upper-level changes in buoyancy. In the tropics the free-tropospheric lapse rate is set by entraining clouds (the zero buoyancy model of Singh and O'Gorman (2013)). Thus, undilute buoyancy increases with warming because of the larger saturation deficit between an undilute parcel and the environment, which scales with saturation specific humidity. As per Seeley et al. (2015), the resulting difference in saturated moist static energy is dominated by the latent enthalpy term at lower altitudes with more water vapor, but by the sensible heat term at higher altitudes. Have the authors considered this?

Reply: The authors have gone through the papers suggested by the honourable reviewer and they also partly agree that the increase in CAPE can be associated with an increase in buoyancy in the lower tropospheric heights due to global warming. However, they have not considered this possibility here as they strongly feel that these hypotheses may not hold very good over tropical Indian regions where overshooting convections are quite common. In addition, the author's feel that the reviewer's suggestion may not provide a strong physical explanation to the obtained trends as they have already been substantiated with firm observational evidences from their previous research attempt.

In that study (Chakraborty et al., 2019 ACP) the authors had utilized multi-station long-term radiosonde profile observations and calculated the average and trend values of CAPE in two separate ways: first from surface till 300 hPa which they called MLCAPE and the second considering the entire cloud column called CAPE. The authors have showed that in average over the Indian region, CAPE is almost double of MLCAPE which shows the ultimate importance of the upper troposphere here. Secondly, in recent years they have obtained a doubling in CAPE while MLCAPE has increased hardly by 40%. So this phenomenon indicates an accelerated strengthening in the UTLS thermodynamics resulting from the ascent in LNB levels to be the dominant factor controlling the net intensification in convective severities over Indian regions. However, later in the study it was found that the observed ascent in LNB strongly corroborates with the ozone depletion related cooling trends (near 150 hPa levels) over the Indian subcontinent. Hence, it was inferred that the rising global warming levels would lead to a positive feedback between convection and UTLS processes which in turn play the most dominant role behind the recent intensification in convective activities over India compared to any

other factors. However, at the end, the authors would like to comment that they will also work on the impacts of lower tropospheric buoyancy on the increasing CAPE both from observation and modelling approaches in their upcoming study.

6. Figure 7 is quite unclear. Please provide units for both y-axes. For the top row, what is "variability"? Please also clarify the meaning of the percentages in the legends in the top row of plots.

Reply: As per reviewer comments, units have now been provided in Figure 7. The long-term trends of both the parameters have been symbolised using the word "variability". Similar to Figure 6, the % values shown in legends depict the approximate 5-yearly linear trends in all these parameters. Here it may be kindly noted that these values are provided just to give a qualitative feel to the readers about the quanta of changes happening in each of these parameters throughout the time span of 150 years. A brief description of the meaning of these % values have now been added in figure captions accordingly.

7. Where do the percentages in Figure 8 come from? I do not see them in the main text. Please also clarify over what time period the changes represent, i.e., last decade minus first decade, last decade minus 2010-2020, 2100-1950, etc.

Reply: As the reviewer has rightly pointed out, the % trends in Figure 8 represents the approximate whole India trends in all the parameters considering the RCP 2.6 and 8.5 scenarios in blue and red, respectively. Further, these trends are considered between two 5-year periods: one at present (2016-2020) and the other at the distant future (2096-2100). Now, an extensive detail on these values has not been added in main text as it is just for the overall representation of final result towards the readers. But as per reviewer's comment a description of these % values has now been added in figure captions.

8. It would probably also be nice to put frequency and radiance units on all pertinent figures (Figures 3, 5, and 7) so that they are not confused with percentages by readers.

Reply: As per reviewer comments, these changes have been incorporated in Figures 3 and 7 of the revised manuscript. However, Figure 5 already looks very congested; hence as suggested, the units have now been mentioned in the figure caption text.

9, Technical Corrections: There are quite a few grammatical few of which are pointed below. pg 7, line 250: descent —> decent & pg 18, line 366: fairs —> fares

Reply: As per reviewer comments, all such grammatical mistakes have now been corrected.

The authors thank the reviewer once again for providing all the suggestions and sincerely accept that these have turned out to be indispensable in pushing and improving the standard of the current work.

---

## Author Comment (AC2)

**Replies to Reviewer #2**

General Comments: The distribution, causative factors and the trends of lightning activities in different parts of India are investigated comprehensively. The main question is that some evidence or more sentences should be added to explain why the author use the variable lightning radiance, and what's the relationship between lighting radiance and intensity of storm clouds. I recommend **minor revision.**

The authors wish to thank the reviewer for the comments/suggestions which significantly improved the content of the manuscript. The authors have addressed all the comments raised by the reviewer and incorporated them in the revised manuscript wherever necessary.

1, The main question is what's the physical meaning of lightning radiance? As I know, in terms of satellite lightning observation, lightning flash rate is most physically significant variable. I'm confused about the relationship between lightning radiance and the intensity of convections, please add some evidence or some more sentences to clarify under what conditions will lightning radiance increase.

Reply: At first the authors want to clarify that lightning radiance represents the amount of charges being neutralized at the ground which in turn also corresponds to the amount of electrical charges generated inside the thundercloud. Now it is well known that satellite observations sense both types of lightning events namely: intercloud and cloud to ground lightning. However, according to previous literature, the former type covers approximately 80-90% of the total events while none of them possess enough charges or energy to shatter the atmospheric insolation to reach the ground. But on the other hand, the remaining 10-20% events have much higher radiance or charges and hence only this type of strikes are capable of reaching the ground and causing enormous socio-economic damage. So, this explains the importance of the lightning radiance. A pertinent discussion on this aspect has now been added in the revised manuscript as follows.

*"While utilizing the lightning radiance measurements from satellite observations, it may be best suited to explain the utility of this data towards the weather and climate as this attribute has never been extensively discussed in past research attempts unlike lightning frequency or flash rate. It is well known that lightning activities originate due to charge separation in mixed phase clouds, but they require a sufficient amount of electrostatic charge to shatter the insolation capacity of the atmosphere and descend to the earth surface further causing widespread damage to life and property. However, a majority of these lightning occurrences are not strong enough and hence they remain as intercloud lightning and hence have no real impact on the climate or socio-economy. On the other hand, according to some novel studies like Uman (1986) only 10-20% of the total lightning activities are strong enough to reach the ground and thereby inflict widespread socio-economic impacts. Thus, the climatological variation of lightning intensity or radiance also needs to be monitored very closely by present policy makers to prevent the chances of any impending catastrophises in future."*

Now as far as the relationship between lightning radiance and its components is concerned, the authors have already cited various research attempts explaining the probable mechanisms responsible for the genesis of lightning. However, for further clarification, the authors would like to recall the non-inductive charging equation from Shi et al. (2015) where the amount of charges created by graupel-ice collisions are expressed in terms of the graupel or ice diameters and their number concentrations along with their relative velocities. The thermodynamic instabilities prevailing inside a thundercloud (denoted by CAPE) represent the relative velocities present between graupel and ice and this in turn impacts the charging rate thus explaining its relationship with radiance. Next, the concentration of graupel and ice particles inside a thundercloud depend on the existing moisture content or TCWV present in the atmosphere and this in turn dictates the number of collisions and hence the radiance associated with lightning. Finally, the diameters of graupel and ice particles determine the collision

area and charge transfer rates responsible for lightning and this in turn is controlled by the aerosol concentrations (AOD). So this explains how lightning radiance is related each of these factors.

$$\left(\frac{\partial Q_{eg}}{\partial t}\right)_{np} = \beta \delta q E_r (1 - E_r)^{-1} \times \frac{1}{\rho_0} |\overline{V}_i - \overline{V}_g| \int_0^\infty \int_0^\infty \frac{\pi}{4} E_{gi} (D_g + D_i)^2 N_g N_i dD_g dD_i$$

Here the Left-hand side represents the charge formation rate, Er refers to the collision and rebounding efficiencies, V indicate the individual vertical velocities for each of the hydrometeors and finally D and N denote the weighted average diameters and number concentrations of both graupel, and ice involved in this process. Now, a detailed explanation on this aspect has already been discussed in the introduction section of the manuscript. Hence this clarification has not been inserted in to the text.

2. If the convection is much stronger, I think both lightning flash rate and lightning radiance should increase, but in this paper, the trends are not inconsistent. I think maybe it's related to different storm clouds. Such as lightning in stratocumulus clouds or thermal convections should be different. Authors can try to give more information about types of clouds in different regions and add more explanations.

Reply: At first, the authors would like to clarify that CAPE or the convective instability is not the major factor controlling both lightning frequency and intensity. Had that been the case, then lightning frequency would not be highest over land while the radiance values are much higher over BoB. Now it is a fact that having too many lightning strikes does not guarantee that all these events will also experience very high intensities (since they do not have an exact one-to-one correlation). Now, a weak lightning event can develop even with moderate instabilities and meagre amounts of hydrometeors; but, in case of intense lightning events a profuse amount of hydrometeor collisions is required (which in turn is not possible without ample moisture supply). Thus, sea regions like BoB always experiences stronger radiance values despite weaker lightning frequency thereby explaining the author's argument.

Now, in view of the reviewer's comments, the authors have gone through sufficient literature survey. A study by Subramanyam (2013) depicted the strongest deep convective clouds over BoB but only during the summer monsoon period. So, to get an overall idea about the impact of clouds, the authors have plotted the yearly averaged high, mid and low-level cloud covers over Indian region. As lightning events require interaction between mixed phase hydrometeors which are mainly found above 5 km; hence high clouds followed by the mid clouds are only expected to show a congruence with lightning.

[Figure]

Figure: Average spatial variations of high, mid and low cloud cover over the Indian Subcontinent

The average spatial variation of cloud types over Indian region also supports this argument showing most frequent high-level clouds over the maritime regions. Also, the amount of high clouds are much higher than the mid-level ones throughout the country thereby implying the stronger role played by the ice clouds behind lightning genesis. However, this relationship fails drastically in a couple of cases. First, the average lightning frequency and radiance values over IGP, WCI or HIM are much higher than PI despite having half of its cloud cover. This facet can be addressed by considering the additional influences of either AOD and CAPE over IGP and WCI or using the concept of orographic lifting in HIM. Also, the contribution of clouds seems relatively useless over AS since it experiences minimum lightning. Hence the authors wish to reaffirm that it is not the clouds alone but a complex combination of CAPE, TCWV and AOD which ultimately determines the lightning properties over the Indian

region. In view of the above and for the sake of simplicity, the authors have decided not to include any discussion about the impact of clouds on lightning in the revised manuscript. However, in future a detailed study can be progressed to investigate the radiative and microphysical impacts of clouds alone on lightning over each Indian subdivision using observations and modelling.

The authors thank the reviewer once again for providing all the suggestions and sincerely accept that these have turned out to be indispensable in pushing and improving the standard of the current work.